

# Identifying and prioritising climate change adaptation actions for greater one-horned rhinoceros (*Rhinoceros unicornis*) conservation in Nepal

Ganesh Pant[1,2], Tek Maraseni[2,3], Armando Apan[2,4] and Benjamin L. Allen[2,5]

[1] Ministry of Forests and Environment, Singhadurbar, Kathmandu, Nepal
[2] University of Southern Queensland, Institute for Life Sciences and the Environment, Toowoomba, Queensland, Australia
[3] University of Sunshine Coast, Sunshine Coast, Queensland, Australia
[4] University of the Philippines Diliman, Institute of Environmental Science and Meteorology, Quezon City, Phillippines
[5] Nelson Mandela University, Centre for African Conservation Ecology, Port Elizabeth, South Africa

Corresponding author
Ganesh Pant, ganeshpant@yahoo.com

## ABSTRACT

Climate change has started impacting species, ecosystems, genetic diversity within species, and ecological interactions and is thus a serious threat to conserving biodiversity globally. In the absence of adequate adaptation measures, biodiversity may continue to decline, and many species will possibly become extinct. Given that global temperature continues to increase, climate change adaptation has emerged as an overarching framework for conservation planning. We identified both ongoing and probable climate change adaptation actions for greater one-horned rhinoceros conservation in Nepal through a combination of literature review, key informant surveys ($n = 53$), focus group discussions ($n = 37$) and expert consultation ($n = 9$), and prioritised the identified adaptation actions through stakeholder consultation ($n = 17$). The majority of key informants (>80%) reported that climate change has been impacting rhinoceros, and more than 65% of them believe that rhinoceros habitat suitability in Nepal has been shifting westwards. Despite these perceived risks, climate change impacts have not been incorporated well into formal conservation planning for rhinoceros. Out of 20 identified adaptation actions under nine adaptation strategies, identifying and protecting climate refugia, restoring the existing habitats through wetland and grassland management, creating artificial highlands in floodplains to provide rhinoceros with refuge during severe floods, and translocating them to other suitable habitats received higher priority. These adaptation actions may contribute to reducing the vulnerability of rhinoceros to the likely impacts of climate change. This study is the first of its kind in Nepal and is expected to provide a guideline to align ongoing conservation measures into climate change adaptation planning for rhinoceros. Further, we emphasise the need to integrating likely climate change impacts while planning for rhinoceros conservation and initiating experimental research and monitoring programs to better inform adaptation planning in the future.

## INTRODUCTION

Climate change is increasingly acknowledged as a critical threat for conserving global biodiversity, which is impacting almost every level of biological diversity including species, ecosystems, ecological interactions, and genetic diversity within species (*Foden et al., 2019*; *IPBES, 2019*). It is triggering changes in phenology, range shifts and species composition (*Chen et al., 2011*; *Rasmussen et al., 2017*; *Haight & Hammill, 2020*). These adverse impacts on biodiversity are likely to intensify in the future, given that the global average temperature is predicted to exceed 1.5 °C by 2100 even under the lowest greenhouse gas emission scenario (*IPCC, 2018*; *Newbold et al., 2020*). Biodiversity continues to decline globally, and many species will possibly become extinct due to the synergetic effects of climate change and land use changes if adequate adaptation measures are not implemented (*Da Silva et al., 2019*; *IPBES, 2019*; *Hannah et al., 2020*).

Climate change adaptation is defined as adjusting to moderate or avoid the harm that is likely to arise from a current or projected change in climate and associated effects (*Smit et al., 2000*). Adaptation priorities of different systems may be different based on the magnitude of change a system has been experiencing or is projected to experience due to climatic stressors (*Watson, Iwamura & Butt, 2013*). Thus, the effectiveness of species conservation strategies relies not only on enhancing knowledge of species and ecosystem responses to these changes but also on envisaging the likely response of humans (*Watson, Iwamura & Butt, 2013*; *Morecroft et al., 2019*). Successful conservation needs to embrace multiple approaches to climate adaptation; however, these are seldom delivered in an integrated way to assist in conservation planning and implementation in the context of the inherent uncertainty associated with future climate conditions (*Smit et al., 2000*). Likewise, the management practices of today may not be relevant under future climate scenarios, and ecologists must go beyond finding the likely climate change impacts and start devising probable solutions (*Hulme, 2005*). In this context, priority should be given to developing adaptation options for the species that are most susceptible to changing climate (*Abrahms et al., 2017*; *Morecroft et al., 2019*).

Greater one-horned rhinoceros (*Rhinoceros unicornis*; hereafter "rhinoceros") is one of the five remaining species of rhinoceros in the world and is currently distributed in a few protected areas in southern Nepal and the northern foothills of India (*Rookmaaker et al., 2016*; *Ellis & Talukdar, 2019*). Rhinoceroses were widespread throughout the Indian subcontinent until the middle of the nineteenth century, but the population sharply declined to only 500 rhinoceros during the 1960s due to poaching and habitat loss (*Rookmaaker et al., 2016*; *Pant et al., 2020b*). However, the rhinoceros population in the wild has been gradually increasing in both India and Nepal over the last two decades following effective conservation initiatives, and the global rhinoceros population at present is more than 3,500 individuals (*DNPWC, 2017*; *Ellis & Talukdar, 2019*). Despite its population recovery from the brink of extinction, rhinoceros is still considered to be at high risk due to poaching and habitat alteration induced by climate change (*Dinerstein, 2003*; *DNPWC, 2017*; *Pant et al., 2020b*). However, the probable impacts of changing climate on rhinoceroses and their habitat have not been well documented (*Pant et al., 2020b*).

Rhinoceros is a habitat specialist and prefers a mosaic of grassland and the riverine forests on alluvial floodplains along the foothills of the Himalayas, where green growth and water remain available throughout the year (*Laurie, 1982*; *Dinerstein & Price, 1991*; *Jnawali, 1995*; *Pradhan et al., 2008*). The insufficiency of suitable habitat is one of the limiting factors for rhinoceros conservation (*Pant et al., 2020b*), and the decline in both quality and quantity of rhinoceros habitat has been documented in rhinoceros-bearing protected areas in both India and Nepal (*Sarma et al., 2009*; *Subedi, 2012*; *Medhi & Saha, 2014*). In Nepal, the rhinoceros population has been gradually shifting westwards, which indicates the change in habitat suitability (*Subedi et al., 2013*) and climate change has been recently acknowledged as an emerging challenge for rhinoceros conservation (*DNPWC, 2017*). The decline in rhinoceros habitat is likely to be intensified in the future due to the impacts of climate change, given that over one-third of the current suitable habitat is predicted to become unsuitable in the next 50 years under the highest greenhouse gas emission scenario (*Pant et al., 2021*).

Over the last few decades, climate change adaptation has been acknowledged as an overarching framework for biodiversity conservation (*Glick, Stein & Edelson, 2011*; *Stein et al., 2013*) and the adaptation actions currently in practice for wildlife management are broadly focused on protected areas, invasive species, ecosystem services, adaptive management, biological corridors, and assisted migration (*LeDee et al., 2021*). There are several examples of adaptation planning for species conservation and ecosystem management from around the globe. For example, national fish, wildlife and plant climate adaptation strategy of the United States (*Burns et al., 2021*), climate change strategy and action plan for Greater Barrier Reef National Park, Australia (*GBRMP, 2012*), climate change adaptation actions for Australian birds (*Garnett et al., 2013*), and climate change adaptation actions for vulnerable seabirds on Albatross Island in Tasmania (*Alderman & Hobday, 2017*) have been formulated. In Nepal, national adaptation plan has been prepared that proposed 11 priority adaptation programs for forests, biodiversity and watershed conservation (*GON, 2021*). However, no specific adaptation actions have been developed to date for particular wildlife species conservation in Nepal.

The aim of this study was to identify, describe and prioritise adaptation actions to moderate the likely effects of climate change on rhinoceros in Nepal. The specific objectives included (1) documenting the ongoing conservation interventions that possibly contribute to climate change adaptation planning, (2) identifying the probable climate change adaptation actions, and (3) guiding the future course of actions to align ongoing conservation measures into adaptation planning. Climate change has been acknowledged as an emerging threat for rhinoceros conservation given that the decline in rhinoceros habitat due to invasive plant species and drying up of wetlands has been documented, and climate-induced hazards including flash floods, prolonged droughts and forest fires are predicted to increase in those areas (*Medhi & Saha, 2014*; *DNPWC, 2017*; *Pant et al., 2020b*; *Pant et al., 2021*). Likewise, *Pant et al. (2020a)* recently reported that rhinoceroses in Nepal are likely to experience a 'moderate' level of climate change vulnerability owing to susceptibility to flash floods, habitat loss due to invasive plant species, increased forest fires and drying up of wetlands due to increased droughts. The findings of the present
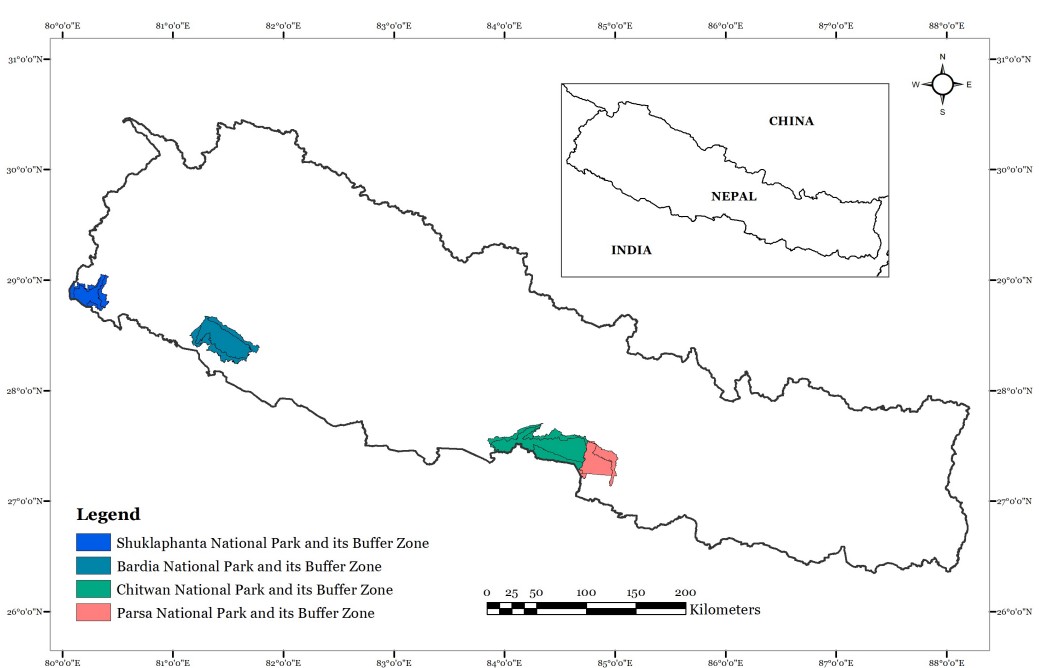

**Figure 1** Location of National Parks (Shuklaphanta, Bardia, Chitwan and Parsa) with extant rhinoceros population in Nepal.

study, if converted into action, are expected to reduce these vulnerabilities to rhinoceros in the era of rapid climate change. Although our focus is on rhinoceros conservation, this study is equally important for adaptation planning for other wildlife species given that rhinoceros is a flagship as well as an umbrella species, its conservation could support in the protection of other naturally co-occurring species (*Roberge & Angelstam, 2004*; *Amin et al., 2006*; *Cédric et al., 2016*).

# MATERIALS & METHODS

## Study area

Nepal extends over 147,516 km$^2$ in South Asia between longitudes of 80°04′ to 88°12′ east and latitudes of 26°22′ to 30°27′ north. We focused our study on all of the protected areas in Nepal with extant rhinoceros populations, namely Shuklaphanta, Bardia, Chitwan and Parsa National Parks, and their surrounding landscapes (Fig. 1). Chitwan National Park (CNP; 95,000 ha) is a stronghold of rhinoceros, and the only source population of rhinoceros in the country (*DNPWC, 2017*). Recently, Parsa National Park (PNP; 62,700 ha) has been colonised by rhinoceros where 3-5 animals have migrated from adjacent CNP (*Acharya & Ram, 2017*). Nearly 100 rhinoceroses were translocated between 1986 and 2017 from CNP to Bardia National Park (BNP; 96,800 ha) and Shuklaphanta National Park (SNP; 30,500 ha) (*DNPWC, 2018*; *Thapa et al., 2013*). Based on the census conducted in 2015 *DNPWC, 2017*, there were 645 rhinoceroses in four National Parks in Nepal, *i.e.,* CNP (605), BNP (29), SNP (8) and PNP (3).

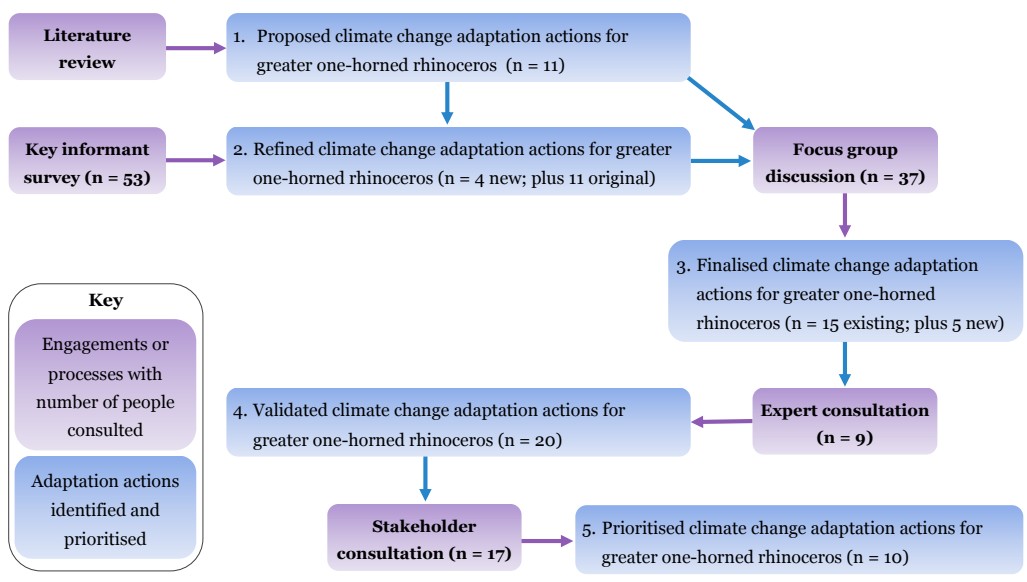

**Figure 2** The methodological approach for identifying and prioritising climate change adaptation actions for rhinoceros conservation in Nepal.

## Methods

This study was conducted with the research permission (075/76 ECO- 2124) from the Department of National Parks and Wildlife Conservation, Nepal and the University of Southern Queensland, Australia has also granted ethical clearance (H19REA001) for the research. We used a combination of literature review, key informant surveys ($n = 53$), focus group discussions ($n = 37$), expert consultation ($n = 9$), and stakeholder consultation for priority ranking ($n = 17$) as methods to identify and prioritise adaptation actions to conserve rhinoceros in the face of climate change (Fig. 2). We collected primary data for this research between February and April 2019. We first developed a set of 11 proposed adaptation actions through a literature review. Later, we refined these actions with inputs from key informants and then finalised a list of 20 adaptation actions through focus group discussions during a stakeholder consultation workshop, where we grouped these actions into nine adaptation strategies. Further, we evaluated and validated the identified adaptation actions through expert consultation. We also documented key informants' insights related to climate change impacts on rhinoceros habitat including the shift in habitat suitability. Finally, we prioritised the identified adaptation actions based on priority ranking by stakeholders and experts.

### Review of relevant literature

Climate change adaptation consists of planned actions aimed at reducing the risks and capitalises on the possible opportunities linked with climate change, which is emerging as a key framework for biodiversity conservation globally (*Füssel, 2007*; *Glick, Stein & Edelson, 2011*). Adaptation planning is regarded as a means to reduce the likely vulnerabilities to climate change and the projected climate scenarios in the future (*Thomas et al., 2019*).

Increasing resilience is an overarching objective of adaptation strategies and principles (*Morecroft et al., 2012*). Decisions on climate change adaptation to biodiversity primarily rely on expert judgement, with supplementary information generated from climate models. This approach also considers managing biodiversity in-situ followed by landscape-level interventions and finally ex-situ conservation through translocation (*Oliver et al., 2012*). Adaptation is characterised by flexible management as a component of well-designed adaptation strategies because of the uncertainties associated with predicted climate change impacts on ecosystems and species (*Glick, Stein & Edelson, 2011*).

Several adaptation approaches are used to incorporate climate change into conservation planning and translating these principles and strategies of climate change adaptation into action. Although various analytical techniques are used for adaptation planning, most of them follow similar steps, including assessing vulnerabilities to the species in relation to the predicted climate change scenarios, determining predicted range shifts for species, identifying promising adaptation options, and then appraising and choosing adaptation actions (*Stein et al., 2013*; *Abrahms et al., 2017*). We followed the participatory adaptation for conservation targets (ACT) framework, as suggested by *Cross et al. (2012)*, which considers the effect of climate change in deciding conservation measures for species, ecosystem and ecological function. This framework is founded on the principle that effective adaptation planning relies predominantly on indigenous knowledge related to ecosystems, and there is no need for detailed forecasts of changing climate or its impacts. We first appraised the generic adaptation actions proposed for biodiversity and wildlife (see *Mawdsley, O'malley & Ojima, 2009*; *Oliver et al., 2012*; *Abrahms et al., 2017*), given that there were no specific adaptation actions already developed for rhinoceros. On the basis of the literature review, including those described in *Pant et al. (2020b)*, we identified 11 adaptation actions relevant to rhinoceros conservation in Nepal.

## Key informant survey

We interviewed 53 key informants in person, including rhinoceros experts, managers of the protected areas, academics, participants from conservation agencies such as the International Union for Conservation of Nature (IUCN), Zoological Society of London (ZSL), World Wide Fund for Nature (WWF), National Trust for Nature Conservation (NTNC), and members of relevant community-based organisations. We purposely selected participants who were directly involved in rhinoceros conservation in Nepal and they were familiar about the ongoing changes in rhinoceros habitat over the years. We documented their understanding of the probable climate change impacts on rhinoceros habitat, and with their input, we identified interventions that are likely to serve as suitable climate change adaptation actions. Five interviewees (9%) were female, and 48 (91%) were male. The fewer number of female interviewees is attributed to the gender imbalance in the biodiversity conservation sector in Nepal. The majority of the participants ($n = 29$; 54%) were government officials and 12 (23%) each from non-government organisations and community organisations. Most of the key informants (>55%) each had 15 years of experience or more in the environmental management sector. These key informants identified four more adaptation actions which were discussed with focus groups.

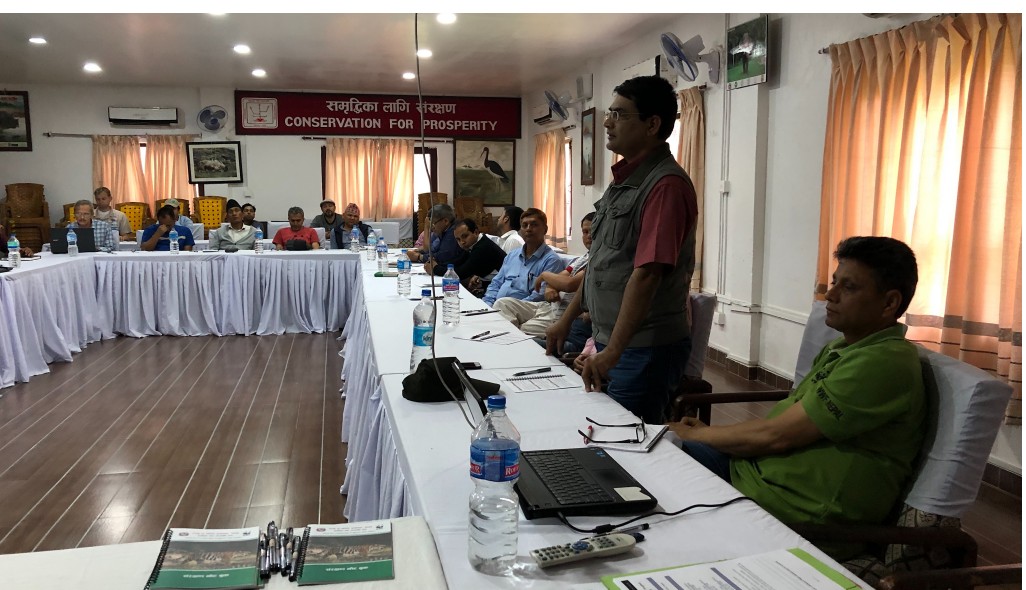

**Figure 3** Participants discussing on climate change vulnerability and adaptation planning for rhinoceros in Nepal.

## Focus group discussion

We conducted focus group discussions on climate change adaptation planning for rhinoceros during a two-day workshop in Chitwan National Park, Nepal on 5-6 April 2019, which was attended by 37 stakeholders representing the department and protected area offices from the government sector, non-governmental organisations, universities and community-based organisations involved in rhinoceros conservation (Fig. 3). The discussion on identifying the adaptation actions was conducted immediately after the vulnerability assessment, the details on assessing climate change vulnerability to rhinoceros in Nepal is presented in *Pant et al. (2020a)*. The information on the existing practices for species-specific adaptation planning and adaptation actions relevant for rhinoceros conservation identified through literature review and key informant survey were provided to the workshop participants. In this session, participants were engaged in a group exercise for identifying the possible adaptation actions, primarily based on the identified climate change vulnerabilities for rhinoceros conservation in Nepal. During the plenary session, each group presented the details of adaptation actions that are expected to reduce the vulnerability of rhinoceros considering predicted climate change impacts, which were then finalised by consensus among all workshop participants. The participants finally agreed on 15 adaptation actions, though five additional potential adaptation actions were added for further discussion with experts.

## Expert consultation

We consulted a cohort of nine experts face-to-face to validate the outcomes of our climate change adaptation focus group exercise for rhinoceros. In doing so, we invited all of the known rhinoceros conservation experts in Nepal from the Department of National

Parks and Wildlife Conservation (DNPWC) and NGOs, including the IUCN, WWF, NTNC and ZSL. Two of the experts were members of the IUCN Asian Rhino Specialist Group. In this face-to-face interaction with experts, adaptation actions identified for rhinoceros conservation were discussed and evaluated. We further prepared a summary report containing the key outcomes of the adaptation planning, which was sent to DNPWC officials and rhinoceros experts for their review and endorsement. Thus, the outcomes of the adaptation workshop were basically validated by nine experts from a range of GOs and NGOs in a series of face-to-face meetings.

## Stakeholder consultation for priority ranking

In a subsequent engagement, we involved key stakeholders having more than ten years of experience in the biodiversity conservation sector in Nepal to assign a rank against each of the 20 adaptation actions on a scale of 0 to 9 (0–Not in priority and 9–highest priority). Out of 23 invitees, 17 stakeholders completed priority ranking individually. Of these 17 participants, 15 (88%) were male, and two (12%) were female. We compiled the assigned ranking score for each of the adaptation actions and calculated the overall score of each adaptation action using the following formula adopted from *Maraseni (2008)*.

$$i = 17, j = 9$$
$$\text{Overall priority score} = \sum (\text{Wi} * \text{Rj})/\text{N}$$
$$i = 1, j = 0$$

where,

Wi = Number of participants selecting a particular adaptation action W (i =1–17) corresponding to a particular rank R (j = 0–9)

Rj = Assigned a rank (j = 0–9) of a particular adaptation action

N = Total number of participants

# RESULTS

## Climate change impacts on rhinoceros and its habitat

The majority of the key informants (>80%) believed that climate change has already started impacting rhinoceroses and their habitat in Nepal (Fig. 4A). Of the 53 key informants, only 6 (9%) had the opinion that the observed changes in rhinoceroses and their habitat dynamics are due to other natural processes over time, though four key informants (7%) were not aware of such changes. Likewise, more than 65% of the key informants considered that rhinoceros habitat suitability in Nepal has been shifting westwards due to climate change (Fig. 4B). However, 11 key informants (20%) felt that the reasons behind this habitat shift were uncertain. Seven key informants (13%) did not know whether there has been a shift in rhinoceros habitat suitability in Nepal or not.

## Climate change adaptation actions for rhinoceros conservation

After reviewing the relevant literatures including *Mawdsley, O'malley & Ojima (2009)*; *Oliver et al. (2012)*, *Watson et al. (2012)*, *Stein et al. (2013)*, *Abrahms et al. (2017)*, we identified a preliminary set of 11 climate change adaptation actions for rhinoceros conservation under nine adaptation strategies that are expected to contribute in reducing
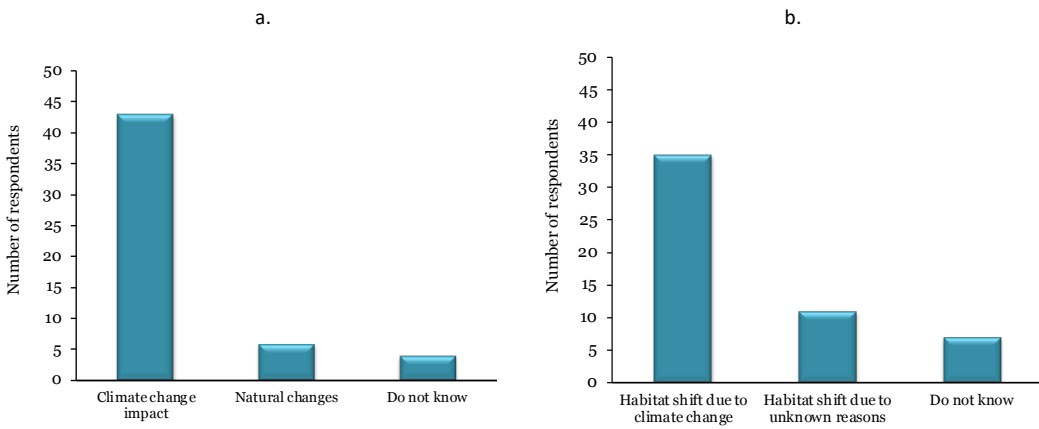

**Figure 4 The perception of key informants about the likely impacts of climate change on rhinoceros habitat in Nepal ($n = 53$).** (A) Key informants' perception of rhinoceros habitat dynamics in Nepal, (B) Key informants' perception on shift in rhinoceros habitat suitability in Nepal.

likely climate change vulnerabilities. These adaptation actions include (i) expanding the existing protected areas, (ii) managing grasslands, (iii) managing wetlands, (iv) controlling invasive species, (v) restoring corridor and connectivity, (vi) conserving biodiversity at the landscape level, (vii) preparing species conservation action plan, (viii) translocating species to other suitable habitats, (ix) strengthening anti-poaching operation, (x) controlling water pollution, and (xi) mitigating human-wildlife conflict. Similarly, four more adaptation actions identified by key informants are (i) establishing new protected areas, (ii) practicing controlled burning, (iii) managing buffer zone, and (iv) conducting periodic census and ID-based monitoring.

In addition, five potential adaptation actions were explored through focus group discussion, which include (i) identifying and protecting climate refugia, (ii) designing and constructing earthen mounds in floodplain grasslands, (iii) integrating climate change impacts in species conservation action plan, (iv) translocating species to future suitable habitats, and (v) initiating experimental research and monitoring of climate change effects. The final set of 20 adaptation actions under nine strategies for rhinoceros conservation in Nepal identified through literature review, key informant survey and focus group discussion, and validated through expert consultation is presented here in Table 1. Of the 20 adaptation actions, 15 (75%) are currently in practice for rhinoceros conservation in Nepal, but these are part of ongoing rhinoceros conservation activities and are not directly linked to climate change.

## Prioritisation of climate change adaptation actions

Out of the 20 identified climate change adaptation actions, ten actions prioritised through stakeholder consultation have been presented in Fig. 5 along with their respective overall score. The adaptation action with an overall score <1 was no longer considered as priority action. Among the others, 'identifying and protecting climate refugia' received the highest

**Table 1  Climate change adaptation actions for rhinoceros conservation in Nepal grouped into different adaptation strategies.** 'Ongoing' refers to the existing conservation interventions that are likely to contribute to increasing the resilience of rhinoceros and ''Probable' refers to the potential adaptation actions for managing rhinoceros in an era of rapid climate change.

| Strategy No. | Adaptation strategy | Adaptation actions | Ongoing | Probable |
|---|---|---|---|---|
| 1 | Increasing the extent of protected areas | a. Expand the existing protected areas | ✓ | |
| | | b. Establish new protected areas | ✓ | |
| | | c. Manage grasslands | ✓ | |
| 2 | Improving management and restoring the existing protected areas | d. Manage wetlands | ✓ | |
| | | e. Practice controlled burning | ✓ | |
| | | f. Control invasive species | ✓ | |
| | | g. Restore corridor and connectivity | ✓ | |
| 3 | Protecting biological corridors, stepping stones and refugia | h. Identify and protect climate refugia | | ✓ |
| | | i. Design and construct earthen mounds in floodplain grasslands | | ✓ |
| 4 | Managing and restoring ecosystem function rather than focusing on specific components | j. Conserve biodiversity at landscape-level | ✓ | |
| 5 | Increasing the matrix by expanding landscape permeability to species movement | k. Manage buffer zone | ✓ | |
| 6 | Focusing conservation resources on species that might become extinct | l. Prepare species conservation action plan | ✓ | |
| | | m. Integrate climate change impacts in species conservation action plan | | ✓ |
| 7 | Translocating species at risk of extinction | n. Translocate species to other suitable habitats | ✓ | |
| | | o. Translocate species to future suitable habitats | | ✓ |
| 8 | Reducing pressures on species from non-climatic sources | p. Strengthen anti-poaching operation | ✓ | |
| | | q. Control water pollution | ✓ | |
| | | r. Mitigate human-wildlife conflict | ✓ | |
| 9 | Evaluating and enhancing monitoring programs | s. Conduct periodic census and ID-based monitoring | ✓ | |
| | | t. Initiate experimental research and monitoring of climate change effects | | ✓ |

priority, with an overall priority score of >6, followed by 'managing wetlands', 'constructing earthen mounds', 'managing grasslands', and 'translocating rhinoceros to suitable areas'.

## DISCUSSION

The result of our study imply that climate change has already started impacting rhinoceros habitat in Nepal. In recent years, climate change has been acknowledged as an emerging threat to rhinoceros (*DNPWC, 2017*). Another study by *Pant et al. (2020a)* has revealed that rhinoceros in Nepal is likely to face a moderate level of vulnerability due to climate change because of severe floods, fragmented habitat, invasive plant species, droughts, small population size and forest fires. We considered these vulnerability factors while identifying the adaptation strategies and actions most likely to enhance its resilience against the impacts

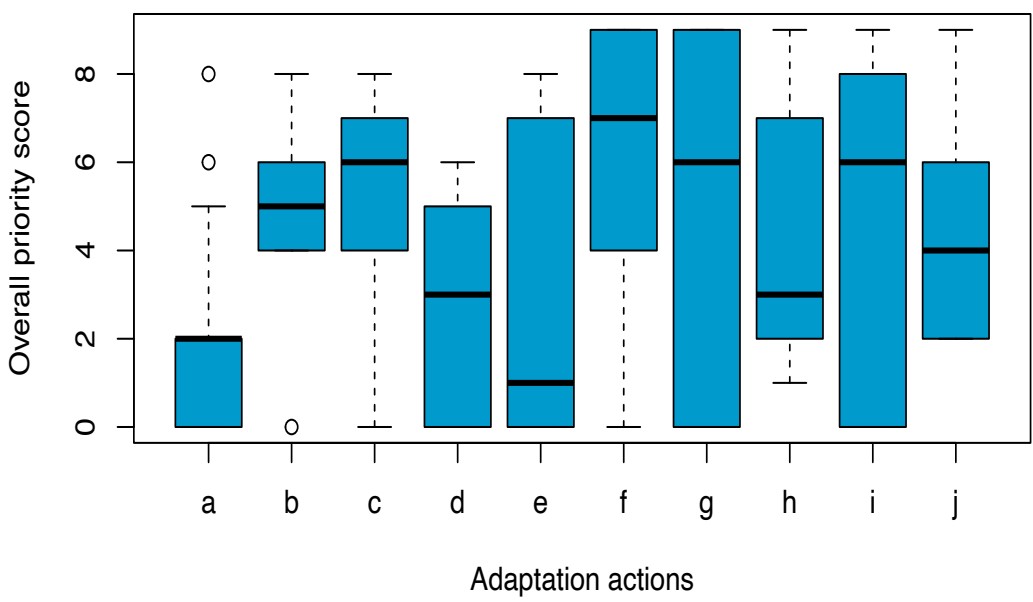

**Figure 5** **The prioritised climate change adaptation actions for greater one-horned rhinoceros conservation in Nepal based on priority ranking by stakeholders (n = 17).** (A) Expand protected areas, (B) Manage grasslands, (C) Manage wetlands, (D) Control invasive species, (E) Restore corridor and connectivity, (F) Identify and protect climate refugia, (G) Design and construct earthen mounds in floodplain grasslands, (H) Develop climate-smart species conservation action plan, (I) Translocate rhinoceros to suitable habitats, (J) Initiate experimental research and monitoring of climate change effectsThe overall priority score '0' denotes least priority and the score '9' is the highest priority.

of climate change. Adaptation strategies and actions need to be revised regularly and should be considered a continual process and not a static endpoint (*Stein et al., 2013*), so our study provides a foundation for the integration of adaptation actions into conservation planning for rhinoceros in Nepal. These findings can be utilised to guide management interventions on the basis of the best information available today and refine these decisions in the future following the principle of adaptive management (*Walsh et al., 2012*).

Those engaged in our study reported a shift in suitable rhinoceros habitat in Nepal, and they considered it a likely climate change impact on rhinoceros. The rhinoceros population has been gradually moving to the western parts of CNP (*Subedi et al., 2013*), and a recent study supports the view that suitable rhinoceros habitat is likely to experience a considerable decrease and shift westwards due to the impacts of climate change (*Pant et al., 2021*). In general, suitable habitat of wildlife species with a moderate level of vulnerability due to climate-induced changes is likely to decline substantially (*Anacker et al., 2013*) but will not be at risk of immediate extinction (*Foden et al., 2019*). Thus, our findings suggest that rhinoceros will have a better chance of persistence through adaptation planning if we can protect both current and future suitable habitat for rhinoceros conservation.

Identifying and protecting climate refugia has been prioritised as one of the most important adaptation actions in this study. Climate refugia, or areas that may serve as a shelter in facilitating the persistence of species amidst climate change impacts are

increasingly acknowledged as an important adaptation strategy (*Morelli et al., 2020*). The increased risk of flooding is an extreme event induced by climate change, which is likely to jeopardise conservation success (*King, 2005*). The entire Terai region is fed by rivers originating in the snow-covered Himalayan mountains, and increasing temperatures lead to increased river flow. Chitwan National Park in Nepal is highly susceptible to this kind of climate-induced flash flooding (*Pant et al., 2020a*). For example, thousands of wild animals were reported dead, including two rhinoceros, during a severe flood episode in August 2017 (*Chitwan National Park, 2017*; *WWF, 2020*). Ten rhinoceros were also swept away through the Indian border and were transported back to the park (*Chitwan National Park, 2017*). In response, a raised soil mound with dimensions of 40 m × 30 m × 2 m was constructed in the buffer zone community forest as an experiment to see whether this type of structure can provide a safe refuge for rhinoceros and other wild animals during severe floods (*WWF, 2020*). We observed the site during our fieldwork in April 2019 and found that the area has been used by rhinoceros and other wild animals, however the effectiveness of these earthen mounds is yet to be evaluated. However, stakeholders and experts believe that such structures could provide safe high grounds for rhinoceros and other animals during flood events. Hence, the construction of earthen mounds in floodplain grasslands was considered to be one potential adaptation action for rhinoceros conservation in Nepal. This strategy is equally important for rhinoceros conservation in India, more specifically in Kaziranga National Park (KNP), given that an estimated 141 rhinoceros have been killed due to severe floods in KNP up until 2019 and 12 rhinoceros were found dead in the recent flood episode of July 2019 alone (*Sharma, 2019*). KNP and the surrounding landscape supports two-thirds of the global population of rhinoceros in the wild (*Pant et al., 2020b*) and the habitat condition and the conservation challenges in Nepal's Chitwan National Park are similar to those in Kaziranga National Park in India (*DNPWC, 2017*; *Puri & Joshi, 2018*).

The findings of our study indicate that improving management and restoring existing protected areas are regarded as essential adaptation strategies for rhinoceros conservation. This could be achieved, in part, through active management of grasslands and wetlands to improve their resilience. Some of the climate change effects in protected landscapes are possible to offset through intensive management of habitat components (*Mitchell et al., 2007*). Grassland management and wetland restoration are key ongoing management activities for rhinoceros conservation in Nepal (*DNPWC, 2017*). Rhinoceros is primarily a grazer and prefers the habitat mosaic of grasslands, riverine forests and wetlands (*Dinerstein, 2003*). But the quality of grasslands in the entire rhinoceros habitat in CNP is degrading due to invasive plants such as *Mikania micarantha* (Murphy et al., 2013). The degradation of wetlands is another serious concern expected to intensify in the future as a result of climate change (*DNPWC, 2017*). Likewise, climate change favours the proliferation of invasive plants (*Hellmann et al., 2008*). Thus, the changes triggered by changing climate should be considered while restoring and maintaining the grassland and wetland habitats to be an effective adaption action for rhinoceros conservation.

Translocation of rhinoceros to other suitable habitat was another prioritised adaption action in our study. Climate change can substantially reduce the availability of suitable
habitat and species with low dispersal capacity will be at higher risk. In such cases, increasing landscape connectivity may not help for dispersal, so translocation of species should be considered as a better option (*Hulme, 2005*). Translocating species to places where they are not present is considered a 'last resort' if unassisted migration to suitable future habitat is very unlikely (*Oliver et al., 2012*). In Nepal, rhinoceroses were only present in CNP during the early 1980s (*Thapa et al., 2013*; *DNPWC, 2017*). To reduce the risk of losing rhinoceros from the likely catastrophic events, poaching and natural calamities, more than 90 rhinoceros were translocated to BNP and SNP between the late 1980s and 2017 (*Thapa et al., 2013*). Habitat suitability models suggest that BNP and SNP are suitable for rhinoceros, and the future suitable habitat is likely to increase (*Pant et al., 2021*). Therefore, continued translocation of rhinoceros to BNP and SNP is a recommended climate change adaptation action for rhinoceros conservation in Nepal.

Expanding protected areas coverage is one of the core strategies for conserving biodiversity, thereby reducing extinction threats (*Dinerstein et al., 2019*). Nepal has made a remarkable achievement in expanding the extent of protected areas (*Acharya et al., 2020*), such that Banke National Park (BaNP; 55,000 ha) and an extended area of PNP (12,800 ha) are recent additions (*DNPWC, 2018*). The extended area of PNP encompasses the suitable habitat of rhinoceros and is currently occupied by rhinoceros (*Acharya & Ram, 2017*). However, BaNP does not have rhinoceros at present and there will be no habitat suitability for rhinoceros in the future either (*Oli et al., 2018*; *Pant et al., 2021*). Thus, increasing the extent of protected areas may not serve as an effective adaptation action if we fail to include suitable habitat for a particular species. In this regard, a few patches of habitat suitable for rhinoceros have been identified in Bara and Rautahat districts to the eastern part of Parsa National Park, which has been used by the rhinoceros straying out from the protected areas (*Acharya & Ram, 2017*; *Rimal et al., 2018*; *Pant et al., 2021*). This area is likely to serve as an additional rhinoceros habitat for protected area expansion. However, further analysis is needed to ensure that poaching and conflict with humans will not jeopardise the conservation of rhinoceros and other wildlife species in those extended areas. Despite being a key adaptation option for biodiversity conservation, stakeholders did not rank the expansion of protected areas in top priority given that only a few patches of potential rhinoceros habitat remain outside the protected areas, >23% of the country is already under protected area system and most of the historical range of the rhinoceros outside protected areas are converted into human settlements (*DNPWC, 2018*; *Pant et al., 2020b*; *Pant et al., 2021*).

This study also acknowledges that corridor connectivity is an integral part of adaptation planning for rhinoceros. Landscape connectivity has also been regarded as a frequently cited adaptation strategy for biodiversity conservation. However, most of the connectivity planning does not directly account for climate-driven range shifts (*Littlefield et al., 2019*). In Nepal, landscape-level conservation has been practised for the last two decades to facilitate the movement of large mammals, including rhinoceros. The forest corridor in western terai between Bardia and Shuklaphanta National Parks is important for rhinoceros conservation given that it connects four rhinoceros-bearing protected areas in a transboundary landscape shared by both India and Nepal that collectively support at least 70 rhinoceros (*Pant et al.,*

*2020b*). Landscape connectivity in this region is vital for rhinoceros conservation given that movement of rhinoceros from one protected area to another has been recorded (*Talukdar & Sinha, 2013*). Maintaining corridors for landscape connectivity can be an important adaptation action for rhinoceros conservation if it accounts for the likely shifts indicated by habitat suitability models.

In practice, it is not possible to develop separate adaptation actions for every wildlife species. However, a number of adaptation actions developed for rhinoceros conservation are expected to benefit other species sharing the same ecosystem given that rhinoceros, like other megaherbivores, require large areas to support viable populations, and their conservation requirements encompass the habitat components required for many other species (*Amin et al., 2006*). For instance, rhinoceros, tiger, and elephant are key wildlife species in Chitwan National Park (*Chitwan National Park, 2013*). Maintaining grasslands and wetlands is a common strategy for conserving these wildlife species given that grassland is a key habitat component for rhinoceros, elephants, and the prey species of the tigers (*Chitwan National Park, 2013*; *Aryal et al., 2016*; *DNPWC, 2017*). In addition, elephants are basically browsers and they require a large volume of fodder and plenty of water for drinking (*Pradhan et al., 2008*). On the other hand, rhinoceros require waterholes for wallowing to regulate their body temperature (*Dinerstein, 2003*). Thus, some of the adaptation actions identified for rhinoceros conservation can serve as adaptation actions for other wildlife species and more specific actions can be further developed based on ecological requirements of these wildlife species occurring in this region.

The implementation of the adaptation actions identified in this study is expected to ensure a greater chance of persistence for rhinoceros well into the future. However, there are a number of factors that are likely to hinder the effective implementation of these adaptation actions for rhinoceros conservation in Nepal. For example, expansion of protected areas and maintaining a functional corridor and connectivity are ideal options for rhinoceros conservation, but very limited suitable habitat for rhinoceros outside protected areas minimises the potential for such intervention (*DNPWC, 2018*; *Pant et al., 2020b*; *Pant et al., 2021*). In this regard, restoring and maintaining the habitat components within protected areas and available biological corridor are among the most feasible options for conserving rhinoceros in the face of likely impacts of climate change that would also help in safeguarding other wildlife species in this region against the adverse impacts of changing climate. Thus, best possible efforts should be made in implementing the adaptation actions, acknowledging that the ideal situation may not be possible for managing large mammals in a human-dominated landscape.

In adaptation planning, uncertainty is regarded as a reality given that many sources of uncertainty exist in ecological processes, including the uncertainties in predicting climate change, possible responses of the species to global warming, and consequences of adaptation actions (*Stein et al., 2013*). Our study, therefore, provides only general guidance in aligning the available adaptation options to adaptation planning for rhinoceros conservation in Nepal. Effective adaptation planning needs to be continually adjusted in such a way that even without having thorough clarity about impacts and consequences, some adaptation options could be implemented and assessed. This approach of 'learning while doing' is

consistent with adaptive management principles (*Gillson et al., 2019*), based on the premise that complete understanding of natural systems is rarely possible, so it is wise to monitor the responses for learning from diversified management interventions (*Williams & Brown, 2016*). Because of its flexible approach and dynamic nature, adaptive management as a fundamental component of adaptation planning should be implemented with as much experimental rigour as possible (*Abrahms et al., 2017*). We expect that the findings of our study will be utilised by protected area managers to make choices based on current information and to refine management actions following an iterative learning process, and we hope that management authorities invest the necessary resources to undertake proper experimental approaches when implementing management activities for rhinoceros conservation.

Adaptation strategies and actions to climate change for other wildlife species in different geographical areas can be formulated following a similar approach, and our research is particularly relevant for Kaziranga National Park in India, where the condition of the habitat and the issues associated with rhinoceros conservation are similar to Chitwan National Park in Nepal (*DNPWC, 2017*; *Puri & Joshi, 2018*; *Ellis & Talukdar, 2019*). Adaptation planning at the species and ecosystem levels are successfully implemented around the world. For instance, *Alderman & Hobday (2017)* developed a set of 24 climate change adaptation actions for vulnerable seabirds on Albatross Island in Tasmania. Likewise, the climate change strategy and action plan for the Great Barrier Reef National Park has been prepared and implemented (*GBRMP, 2012*). Such climate change adaption strategies and actions for wildlife species have not yet been formulated in Nepal. Our study is the first of its kind in Nepal and is expected to assist a vulnerable species to withstand the likely negative impacts of climate change. We focused on a single species given that the nature and degree of the impacts associated with changing climate are species-specific, even amongst closely related species. For example, two species of rhinoceros were affected differently by climate change in Kruger National Park –while births decreased and mortality increased for white rhinoceros, there were no such impacts on black rhinoceros due to the recent severe drought events (*Ferreira, Roex & Greaver, 2019*).

## CONCLUSIONS

This study has identified, shortlisted, selected and ranked a suite of 20 plausible adaptation actions under nine adaptation strategies that are expected to enhance the resilience of rhinoceros to the likely adverse impacts of climate change. Of these, 75% of adaptation actions are already being implemented. However, these actions are implemented in different contexts without explicitly assessing the likely climate change impacts on the species and its habitat. Based on our findings on identifying and prioritising adaptation actions and analysis of the results from vulnerability assessment (*Pant et al., 2020a*), we recommend the following conservation interventions for effective climate change adaptation planning for rhinoceros in Nepal:

a.  Protect identified climate refugia for rhinoceros conservation, particularly in western Nepal around Bardia and Shuklaphanta National Parks and further evaluate the

habitats that are likely to become suitable for rhinoceros in the future, aiming to prioritise and spatially integrate these climate refugia. The priority should be given to restore biological corridors and maintain landscape connectivity to facilitate natural dispersal of rhinoceros between suitable habitats.

b. Identify areas in floodplain grasslands with the help of comprehensive flood modelling to create elevated refuges for rhinoceros during climate-induced flood episodes. This is particularly relevant for rhinoceros conservation in Chitwan National Park, which is highly susceptible to heavy rainfall and flash flooding.

c. Improve and restore the existing protected areas through active management of grasslands and wetlands including controlled burning, and invasive plant species control. This is particularly important in Chitwan National Park, which is likely to experience more climate-induced habitat alteration.

d. Translocate rescued rhinoceros to other suitable areas in the future. Where rescues are required, serious consideration should be given to releasing rescued rhinoceros into Bardia and Shuklaphanta National Parks rather than bringing them back to Chitwan National Park.

e. Increase the extent of protected areas, by either creating new protected areas or expanding existing ones. Priority should be given to including forest patches in Bara and Rautahat districts to the eastern part of Parsa National Park which is likely to serve as an additional habitat for rhinoceros conservation.

f. Revise the conservation action plan developed for rhinoceros conservation in Nepal, integrating the identified climate change adaptation actions that are expected to reduce the likely vulnerabilities to rhinoceros due to climate change.

g. Initiate experimental research related to aspects of rhinoceros ecology with the best chance of informing future climate change adaptation planning. This is expected to provide better insights on the likely consequences of climate change so it can be utilised in refining adaptation actions in the future following adaptive management principles.

## ACKNOWLEDGEMENTS

First author thanks the Government of Nepal for providing study leave and the Government of Australia for offering the Endeavour Scholarships. We thank Department of National Parks and Wildlife Conservation, Nepal for the support to conduct this research. We appreciate experts and stakeholders for their active participation in the surveys, group discussions and consultations. We acknowledge scholars from Nepal and abroad for their constructive feedback to refine this paper. Finally, we thank Dr Barbara Harmes, English Language Advisor at the University of Southern Queensland, Australia, for her editorial support.

### Funding

This study was supported by a travel grant from the Graduate Research School of the University of Southern Queensland and a student research grant from USAID funded

Hariyo Ban Program/WWF Nepal. The funders had no role in study design, data collection and analysis, decision to publish, or preparation of the manuscript.

**Grant Disclosures**

The following grant information was disclosed by the authors:

The Graduate Research School of the University of Southern Queensland.

USAID funded Hariyo Ban Program/WWF Nepal.

**Competing Interests**

The authors declare there are no competing interests.

**Author Contributions**

- Ganesh Pant conceived and designed the experiments, performed the experiments, analyzed the data, prepared figures and/or tables, authored or reviewed drafts of the paper, and approved the final draft.
- Tek Maraseni, Armando Apan and Benjamin L. Allen conceived and designed the experiments, authored or reviewed drafts of the paper, and approved the final draft.

**Human Ethics**

The following information was supplied relating to ethical approvals (i.e., approving body and any reference numbers):

The University of Southern Queensland granted ethical clearance (Ethical Application Ref: H19REA001).

**Ethics**

The following information was supplied relating to ethical approvals (i.e., approving body and any reference numbers):

Department of National Parks and Wildlife Conservation, Nepal (Research Permission: 075/76 ECO- 2124).

**Data Availability**

Raw data and analysis are available in the Supplemental Files.

**Supplemental Information**

Supplemental information for this article can be found online at http://dx.doi.org/10.7717/peerj.12795#supplemental-information.

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
