# Peer review of "Identifying and prioritising climate change adaptation actions for greater one-horned rhinoceros (Rhinoceros unicornis) conservation in Nepal"

_PeerJ, doi:10.7717/peerj.12795_

## Round 0.1 · original submission · Major Revisions

I think the authors have done a good job, however, the reviewers suggest rewriting some sections to improve the writing. The results should be rearranged, some of the conclusions should be moved to results, etc. Please follow the suggestions of Reviewers 1 and 3 and please clarify that many times the ideal is not the possible when dealing with overpopulated countries and large animals such as this case. Also, Reviewer 3 is very critical with some points, read them carefully and answer in detail why or why they are not taken into account. I hope you can make these changes soon to re-evaluate your submission.

Reviewer 1 ·

Basic reporting

English writing is excellent. Research is well designed with a clear structure and findings. Detailed are in the additional comments section.

Experimental design

Well designed. Detailed are in the additional comments section.

Validity of the findings

The findings are valid. Details are in the additional comments sections.

Additional comments

Comments:
Under reviewed manuscript aims to identify and prioritize climate change adaption actions for greater one-horned rhinoceros conservation in Nepal. The authors have done a good job of exploring adaptions actions and provided a robust discussion, which is directly relevant to the policy planning implantation. I have the following comments for the improvement of this manuscript.

Introduction
The authors have done a fair job of describing the situation. However, I see a need for one paragraph that describes ‘existing knowledge’ from other countries (assuming that this is novel for rhinoceros in Nepal) or for other animals. Please add the relevant literature on it. This paragraph can be a second last paragraph just before the one starts at page number 88.

Also, it may be worth pointing how strategies may be similar or different between herbivores, carnivores, and Omnivores. From a management standpoint, it is nearly impossible to make unique strategies for each animal. For example, in the Chitwan National Park, elephants, rhinos, and Tigers are the three majestic large animals. Introducing this reality in the introduction section and later elaboration on the discussion section may be of interest to broader protected area management readership.

Results:

The results section needs major revamping. Why you are only describing the findings from key informant analysis? As of now, the half-page results section is very confusing as it is a hodgepodge of everything. Please make subsections or dedicate at least 'one paragraph result' for each of the 5 methods that were used for data collection. For example, describe what did you find from 1) literature review, 2) key informant survey, 3) focus group discussion, 4) expert consultation, and 5) stakeholder consultation. If the findings from the literature review are difficult to elaborate on, please delete them from the method section. Indeed, this verbiage can be safely removed from here and put in the introduction section to identify the knowledge gap.

Discussion/conclusion:

I think verbiage in paragraphs between 285-303 can go to the results section with some modifications. This sounds like the finding coming from the literature review.

The information in lines 305-324 is highly management relevant. It may be worthwhile to provide similar information from neighboring Indian national parks as there was flooding on the Indian side as well. This information will be of interest to park personnel from India.

I also suggest acknowledging that some of the adaption actions are difficult to implement. These protected areas are bottlenecked between large human settlements where the majority of Nepali citizens reside. While the expansion of existing protected areas be an ideal adaption action, it is not likely to be implemented in a heavily populated country. While I appreciate the conservation mindset of the authors, I simply do not see species-specific protected area expansion being practically possible in the heavily populated Terai region of Nepal. The fact is that people vote to elect lawmakers (animals don’t) and resettlement is a practically very difficult task. Please acknowledge this reality of the country in the discussion section. In doing so, please add a paragraph highlighting what adoption actions could be universally useful for most mega animals in Nepal. As you have pointed, Banke National Park is not suitable for rhinoceros habitat but it is suitable for Tiger habitat. Tiger habitat is also being impacted by climate change. Therefore, broad-based climate adoption actions should receive the highest priority. The added verbiage will add on to make this outlet a management-relevant article.

With these comments withstanding, the authors have done a good job in discussion and the detailed management recommendations. Overall, I enjoyed reading the article.

Reviewer 2 ·

Basic reporting

The paper is very clear and highlights the predicament that conservation management has to confront in helping species adapt to climate change. An evaluation of the adaptation options helps conservation management prioritise key ones to focus on and often enables action through the various methods of engagement employed in the study. Excellent use of language throughout the article. The introduction and abstract does justice to the rationale beefed up by sufficient relevant literature.

The article flow and structure is adequate and simple for the reader to follow with solicited knowledge from the various methods demonstrating that Climate change impacts on rhinoceros have already started impacting their habitat in Nepal and that the shift in habitat suitability has been shifting westward. Climate change adaptation actions most relevant for rhinoceros conservation in Nepal is discussed and ten actions that are of top priority have been presented.

The raw data and analysis process has been shared and is adequate to arrive at the conclusions of the study.

Experimental design

The experiments design is sound and clearly articulated. The area of species climate change adaptation requires interventions from conservation management and as such very few studies exist that captures the process of gearing them towards future interventions. A review of the key adaptation actions is solicited and conservation management buy-in is crucial. The methods are detailed and repeatable.

Validity of the findings

I am satisfied with the manuscript as findings are supported by statistically robust analysis. The discussion of each adaptation option is robust leading to practical recommendations.

Additional comments

No additional comment.

·

Excellent Review

This review has been rated excellent by staff (in the top 15% of reviews)
EDITOR COMMENT
Thank you very much for having made such a meticulous and detailed review of this paper. I think your contributions have been key to better quality and as you wrote “to provide the reader with a better understanding of the context and also to ensure the results can be used in a more meaningful way”. The quality and clarity of this type of work is key, especially when investigating key issues such as adaptations of threatened species to climate change. Thank you very much for your effort.

Basic reporting

This paper investigates the very important topic of climate change adaptation for threatened species, in this case, the greater one-horned rhinoceros. While the concept is sound, and the paper is well written, the paper presents no new information and is not formulated in a particularly helpful way. The paper details a series of consultations on one-horned rhino climate change vulnerability and potential adaptation options. Most people consulted believe the species to be climate change vulnerable. However, though the authors mention presenting vulnerability assessment methods at their expert workshops, no details of how rhino climate change vulnerability had been assessed are presented. It is unclear to what extent the rhinos being investigated are believed to be vulnerable or why. The results do show that workshop participants believe the animals to be climate vulnerable. However, there is no basis for these conclusions and no details of the mechanism of vulnerability are provided. There are a number of papers detailing how climate change vulnerability assessment for species should be conducted (e.g. Foden et al. 2019). The method followed should be referenced and details provided.

Adaptation options were discussed at these workshops and some very generic adaptation responses are ranked. The discussion then covers some of the adaptation responses in more detail. A large emphasis is placed on the experience of people consulted and the multiple steps in the process, but very little on what is known of climate change impacts on rhinos and what experts think can meaningfully be done in this regard in the country of interest.

I would suggest that a complete restructuring is required to provide the reader with a better understanding of the context and also to ensure the results can be used in a more meaningful way. I would rework and present it as follows:

First provide some background on the species: what habitats does it occur in, what climate-related changes are expected or being observed in these areas and why is the species believed to be climate-vulnerable? There appears to have been a study in this regard (Pant et al. 2020 is cited), so the key findings should be detailed along with the projected climate changes for the area. For example, in the discussion, it appears that higher temperatures are linked to snowmelt, which may cause flooding in some areas – where is this a problem, and how frequently are such floods predicted?
Linking aspects of vulnerability to a framework such as the one in Foden et al. 2019 may be useful in this regard.
[In terms of methods, also detail which of this information was provided to the workshop participants?]

The results should focus on each of the adaptation options presented and why they might be useful to rhinos in this region. Which adaptation options speak to which vulnerabilities? For example, the two key vulnerabilities that emerge in the discussion appear to be susceptibility to flash floods and loss of habitat to bush encroachment (not a climate change effect, but driven by CO2). Linking the mechanism of vulnerability with particular adaptations can assist in prioritizing and tailoring such adaptation locally.

The discussion can then focus on detailing a location and species-specific adaptation plan based on the identified priorities. How can the highest priority actions be integrated spatially and why did experts not rate some options (e.g. protected area expansion) as a high priority – i.e. local constraints. As it stands, the generic adaptation options we all know about are just rehashed, with one or two weak links to the local context.

Reading between the lines, I would imagine that adaptation would include construction of flood refugia in floodplains (detail which areas these are); Use of prescribed burning and alien clearing to maintain preferred habitat; metapopulation management such that animals are transported from high-risk areas (which are these) to areas of lower risk (where are these and what makes these areas lower risk)? All of this needs to be contextualized in terms of local predictions of climate change.

Specific comments
Line 52 – there is a new IPCC report out. If possible cite the latest figures
Methods – Figures 4-6 show results and should not be referred to in the methods.

Figure 4 and 5 could probably be combined. Also, for figure 4 – the way this is phrased assumes that changes in rhino habitat have taken place – was no change not an option? Nothing up to this point indicated that a change should have been expected? The population had recovered, but poaching and general land degradation remain a threat. One might presume that these factors could also impact on the rhinos. The way that the question is phrased seems odd when no evidence of change has been presented.

Figure 6 – explain in the caption what 0 versus 8 on the priority scale means – which is high?

Focus group discussions
Lines 190-191 Which methods were explained for assessing species climate change vulnerability?
How was exposure (current and future) to climate change evaluated or understood by workshops participants?

Acronyms should be defined. Some like IUCN and WWF are well known, but others like DNPWC and NTNC are not.

Lines 198-211 (Expert consultation) could be summarized to “The outcomes of the adaptation workshop were validated by 9 experts from a range of NGOs in a series of face-to-face meetings.”
Results - It is not clear on what basis people believe changes in habitat have been due to climate change. The researchers have not even detailed that changes in habitat have occurred. Has this been measured? Over what time period? There are a huge number of unknowns as well as drivers of habitat change, so asking these questions of experts seems unfair and preemptive/leading.

Line 261 – no evidence of climate change impact on rhinos is presented whatsoever. In what way have experts and stakeholders deemed rhinos to be vulnerable to climate change? Which aspects of climate change are of concern?

Identifying broad classes of adaptation is not useful. Aside from 3i, Table 1 lists key generic climate change responses that are very well known and adds nothing new to show what can be done for this species in this area. Where are new protected areas needed? What are the priority areas for corridors – can these species even be expected to move across these large areas? In what way can wetland, fire and grassland management be used to aid rhinos? What type of habitats and fire regimes are required?

275 – what constitutes good rhino habitat and is climate change expected to change the distribution of such habitat westwards? On what grounds are experts making these decisions?

Lines 305-318 – this sort of information (details of mounds built to provide high ground in the case of flooding) would be useful upfront in the introduction, then the use and application of this form of adaptation in the study region can be detailed in the results and discussion

Paragraph beginning 326 – impacts of climate change and CO2 are in fact not one and the same, therefore bush encroachment or similar cannot be said to be driven by climate change. Be careful of phrasing. This paragraph gives the impression that it is not climate change so much as alien species and bush encroachment that may be threatening the rhino. The required responses would therefore be different, compared to concerns of more frequent extreme weather events or rising temperatures.

Line 355 – Are there areas in Nepal currently not protected that do contain suitable habitat for the species? This is the key question to ask when considering protected area expansion as an adaptation option.

Paragraph beginning 357 – are the protected areas fenced?

The conclusions are currently a repeat of table 1 and figure 6. To be more informative, these should be presented in order of perceived importance.

Experimental design

It is not clear how the expert workshops were run and what information was presented. This is key to informing what the experts indicated in terms of climate change adaptation.

Validity of the findings

I am sure that all the adaptations identified would be useful for rhinos in the face of climate change. There is however not enough detail in the introduction and methods on the type of threats faced and the information provided to experts to come to these conclusions.

---

## Round 0.2 · Minor Revisions

We have sent your manuscript to one of the reviewers for review again. Consider that the manuscript is almost ready for publication and requires only a few minor clarifications that I trust will be able to make shortly,in order to accept your paper for publication.

·

Basic reporting

The authors’ revisions have made the manuscript much clearer. The examples added in the discussion are particularly helpful. The process flow of developing the recommended adaptation options is, however, still a little unclear, but easily rectifiable. Updating Figure 2 would go a long way to correcting this. Several rounds of engagements were held with different groups of stakeholders and experts, each producing adaptation options and/or prioritizing them. How these options feed into subsequent steps is however unclear. As a first step, I would include the number of people involved at each stage and the number of actions identified in the boxes within Figure 2 (refer to the attached example). Secondly, how the actions produced at each step flow into subsequent stages needs to be made more explicit through the use of arrows.

For example, in the initial stage, it appears that 11 actions are suggested. At the refining stage, four are proposed, which appear to be in addition to the 11 from the initial stage – I assume all 11 were considered and supported? The finalizing stage appears to have 15 + 5 options – presumably, the 15 are the 11 plus the 4, but this is unclear? The text suggests that the four options from the key informant surveys fed directly into the stakeholder discussions, but what about the steps in between? – I think it’s just the text that is misleading – see comments on the document itself.

With regards to the figure, I would almost suggest removing the boxes on the left (initial through to prioritizing phase) and just showing how different engagements and sets of adaptation options feed into one another using appropriate arrows. I have included an example of the type of diagram you might want to use (it may require further edits and updates) within the manuscript PDF I am attaching.

It is not clear in the last step of stakeholder engagement why only 10 options are prioritized. Is this because some were grouped together or because some were not prioritized and therefore no longer considered? If some were dropped off, how was the cut-off decided? Perhaps update text and figure to make this process clearer.

Other comments:
I have made a number of minor text corrections and suggestions on the document itself, including the removal of duplicated text.

Experimental design

See above comments on edits to figure 2.

Validity of the findings

No further comments

---

## Round 0.3 · accepted · Accept

Dear authors, I believe that in this version you have included all the suggestions so that your paper is ready to be accepted for publication.